# Theories and Analysis of Functionally Graded Beams

**J. N. Reddy [1,\*], Eugenio Ruocco [2], Jose A. Loya [3] and Ana M. A. Neves [4,\*]**

[1] J. Mike Walker '66 Department of Mechanical Engineering, Texas A&M University, College Station, TX 77843-3123, USA

[2] Department of Engineering, University of Campania "Luigi Vanvitelli", Via Roma 29, 81031 Aversa, Italy; eugenio.ruocco@unicampania.it

[3] Department of Continuum Mechanics and Structural Analysis, University Carlos III of Madrid, 28911 Madrid, Spain; jloya@ing.uc3m.es

[4] Department of Mechanical Engineering, Faculty of Engineering, University of Porto, 4200-465 Porto, Portugal

\* Correspondence: jnreddy@tamu.edu (J.N.R.); ananeves@fe.up.pt (A.M.A.N.)

**Abstract:** This is a review paper containing the governing equations and analytical solutions of the classical and shear deformation theories of functionally graded straight beams. The classical, first-order, and third-order shear deformation theories account for through-thickness variation of two-constituent functionally graded material, modified couple stress (i.e., strain gradient), and the von Kármán nonlinearity. Analytical solutions for bending of the linear theories, some of which are not readily available in the literature, are included to show the influence of the material variation, boundary conditions, and loads.

**Keywords:** analytical solutions; beams; classical theory; shear deformation theories; functionally graded structures; modified couple stress; numerical results

## 1. Introduction

### 1.1. Preliminary Comments

*Beams* are structural members that have a ratio of length-to-height that is very large (say, $a/h > 10$) and are subjected to forces both in-plane and transverse to the plane that tend to bend about an axis perpendicular to their length. Such members are known as structural elements and their study constitutes *structural mechanics*, which is a subset of solid mechanics. Due to the particular structural configuration (i.e., one dimension being much bigger in comparison to the cross-sectional dimensions), the deformation and stress fields can be predicted, for most practical engineering problems, with structural theories known as the *beam theories*. Beam theories are derived from the three-dimensional elasticity theory by making certain simplifying assumptions concerning the deformation (kinematics) and stress states. The development of such theories dates back to Galileo Galilei, Leonardo da Vinci, and Jacob Bernoulli and Euler. The first one is *the Euler–Bernoulli beam theory*, in which the transverse shear strain is neglected, making the beam infinitely rigid in the transverse direction. The second one that accounts for the transverse shear strain ($\gamma_{xz}$) is popularly known as *the Timoshenko beam theory* [1,2].

All modern developments are refinements to the above stated two theories, where the displacement fields are expanded in terms of powers of the thickness [3] and accounting for other non-classical continuum mechanics aspects (e.g., stress and strain gradient effects and material length scales). For example, a general higher-order theory is of the form

$$\mathbf{u} = u_x\,\hat{\mathbf{e}}_x + u_y\,\hat{\mathbf{e}}_y + u_z\,\hat{\mathbf{e}}_z \tag{1}$$

where

$$u_x(x,z) = \sum_{i=0}^{m} z^i \phi_x^{(i)}(x), \quad u_y = 0, \quad u_z(x,z) = \sum_{i=0}^{p} z^i \psi_z^{(i)}(x) \tag{2}$$

where $\phi_x^{(0)} = u$ and $\psi_z^{(0)} = w$ are the midplane displacements along the $x$ and $z$ directions, respectively, and $\phi_x^{(i)}$ and $\psi_x^{(i)}$ can be mathematically interpreted as higher-order generalized displacements with the meaning

$$\phi_x^{(i)} = \frac{1}{(i)!}\left(\frac{\partial^i u_x}{\partial z^i}\right)_{z=0}, \qquad \psi_z^{(i)} = \frac{1}{(i)!}\left(\frac{\partial^i u_z}{\partial z^i}\right)_{z=0} \tag{3}$$

For a general third-order beam theory, we have $m = 3$ and $p = 2$ in Equation (2). The third-order beam theory of Reddy, derived from the third-order plate theory (see Reddy [4–7] and Heyliger and Reddy [8]), adopts a displacement field that is a special case of Equation (1) and imposes zero transverse shear stress conditions on the bounding planes (i.e., top and bottom faces) of the beam to express the variables introduced with the higher order terms in terms of the variables appearing in the Euler–Bernoulli and Timoshenko beam theories.

### 1.2. Functionally Graded Structures

#### 1.2.1. Background

Functionally gradient materials (FGM) are a class of composites that have a gradual variation of material properties from one surface to another. These novel materials were proposed as thermal barrier materials for applications in space structures, nuclear reactors, turbine rotors, flywheels, and gears, to name only a few. In general, all the multi-phase materials, in which the material properties are varied gradually in a predetermined manner, fall into the category of functionally gradient materials. Such property enhancements endow FGMs with material properties such as the resilience to fracture.

In the last two decades, a large number journal papers dealing with functionally graded beams and plates have appeared in the literature and a critical review of these papers is not a focus of this introduction to FGM structures (see, e.g., Birman [9] and Klusemann [10] for a review). The works of Praveen and Reddy [11] also considered von Kármán nonlinearity in functionally graded plates.

#### 1.2.2. FGM Material Models

A typical FGM represents a particulate composite with a prescribed distribution of volume fractions of constituent phases. In the case of beams, the material properties are assumed to vary continuously through the beam height. Several models are available in the literature, but the Voigt (or power-law) and Mori–Tanaka schemes [12] have been generally used for the study of FGM structures.

The advantage of the Voigt scheme is the simplicity of implementation and the ease of computation. According to the Voigt scheme, the effective properties are the arithmetic average of constituent property ($P$) and are given by (see, e.g., [11,13])

$$P(z) = (P_1 - P_2)f(z) + P_2, \quad f(z) = \left(\frac{1}{2} + \frac{z}{h}\right)^n \tag{4}$$

### 1.3. Modified Couple Stress Effects

#### 1.3.1. Background

Theories that account for microstructural length scales are the modified couple stress theory of Mindlin [14], Koiter [15], and Toupin [16] and the strain gradient theory of [17–19]. A more complete review of the early developments can be found in the work of Srinivasa and Reddy [20]. The strain gradient theory is a more general form of the modified couple stress theory and the relationship between the modified couple stress theory and the strain gradient theory can be found in the recent work of Reddy and Srinivasa [21]. In recent years a number of attempts have been made to bring microstructural length scales into the continuum description of beams and plates. Such models are useful in determining the structural response of micro and nano devices made of a variety of new materials that require the consideration of small material length scales over which the neighboring

secondary constituents interact, especially when the spatial resolution is comparable to the size of the secondary constituents.

Microstructure-dependent theories are developed for the Bernoulli-Euler beam by Park and Gao [22], for the shear deformable beams and plates by Ma, Gao, and Reddy [23,24], and for vibration and buckling of shear deformable beams by Araujo dos Santos and Reddy [25–27]. In the last two decades, Reddy and his colleagues [23–30] have published a large number of papers dealing with linear and nonlinear bending of classical and first- and third-order shear deformable beams using the modified couple stress theory. Some of these works have accounted for the von Kármán nonlinearity and functionally graded materials. Of course, there are many papers by other colleagues on the same topics, which are not cited here and references to them can be found in the works already cited here. The von Kármán nonlinearity may have significant contribution to the response of beam-like elements used in micro- and nano-scale devices such as biosensors and atomic force microscopes (see, e.g., Li et al. [31] and Pei et al. [32]).

### 1.3.2. The Strain Energy Functional

The modified couple stress theory is based on the hypothesis that the rate of change of macrorotations cause additional stresses, called *couple stresses*, in the continuum. The rate of change of rotation is represented by the curvature tensor $\chi$, which is defined by

$$\chi = \frac{1}{2}\left[\boldsymbol{\nabla}\boldsymbol{\omega} + (\boldsymbol{\nabla}\boldsymbol{\omega})^{\mathrm{T}}\right] \tag{5}$$

where $\boldsymbol{\omega}$ is the rotation vector

$$\boldsymbol{\omega} = \frac{1}{2}\boldsymbol{\nabla} \times \mathbf{u} \tag{6}$$

and $\mathbf{u}$ is the displacement vector of an arbitrary point in the beam. Physically, $\boldsymbol{\omega}$ denotes the macrorotation at a point of the continuum.

According to the modified couple stress theory [18], the strain energy potential of an elastic beam can be expressed as

$$U = \frac{1}{2}\int_0^L\left[\int_A (\boldsymbol{\sigma} : \boldsymbol{\varepsilon} + \mathbf{m} : \chi)dA\right]dx \tag{7}$$

where $L$ is the length of the beam, $\boldsymbol{\sigma}$ is the Cauchy stress tensor, $\boldsymbol{\varepsilon}$ is the simplified Green–Lagrange strain tensor, and $\mathbf{m}$ is the deviatoric part of the symmetric couple stress tensor. In the coming sections, these relations will be specialized to various beam theories. The couple stress tensor $\mathbf{m}$ is related to the curvature tensor $\chi$ through the constitutive relation [14]:

$$\mathbf{m} = 2G\ell^2\chi \tag{8}$$

where $\ell$ is the length scale parameterand $G$ is the shear modulus. As pointed out in [33] the material length scale parameter of the modified couple stress theory is not constant for an especial material and changes as the size of a structure changes. To determine this value, experimental data for all different sizes are required.

The present paper outlines the displacement fields of the three theories (classical, first-order, and third-order), the governing equations, and analytical solutions of straight beams for the linear case. To keep the size of the paper within reasonable limits, many details are not included here, and interested readers may consult the forthcoming book by the senior author on beams and circular plates [34], which is very comprehensive in its treatment of the theories, analytical solutions by exact means, the Navier solution approach, and numerical solutions by variational and finite element methods.

## 2. Classical Theory of Beams (CBT)

### 2.1. Kinematics

The displacement field of the classical beam theory (CBT) is constructed assuming that transverse lines perpendicular to the beam axis ($x$) remain: (1) straight, (2) inextensible, and (3) perpendicular to the tangents of the deflected $x$-axis. These assumptions, known as the Euler–Bernoulli hypothesis, result in the following displacement field:

$$\mathbf{u}(x,z) = [u(x) + z\theta_x(x)]\hat{\mathbf{e}}_x + w(x)\hat{\mathbf{e}}_z, \;\; \theta_x \equiv -\frac{dw}{dx}, \tag{9}$$

where $(\hat{\mathbf{e}}_x, \hat{\mathbf{e}}_z)$ are the unit basis vectors along the $(x,y)$ coordinates and $(u, w)$ denote the axial and transverse displacements, respectively, of a point on the midplane of the beam.

Based on the displacement field in Equation (9), the only nonzero strain in the present case is (see Reddy [35]) $\varepsilon_{xx}$:

$$\varepsilon_{xx}(x,z) = \frac{du}{dx} + \frac{1}{2}\left(\frac{dw}{dx}\right)^2 + z\left(-\frac{d^2w}{dx^2}\right) \equiv \varepsilon_{xx}^{(0)} + z\,\varepsilon_{xx}^{(0)}, \tag{10}$$

$$\varepsilon_{xx}^{(0)}(x) = \frac{du}{dx} + \frac{1}{2}\left(\frac{dw}{dx}\right)^2, \;\; \varepsilon_{xx}^{(1)}(x) = -\frac{d^2w}{dx^2}. \tag{11}$$

The only nonzero components of the rotation and curvature are

$$\omega_y = \frac{1}{2}\left(\frac{\partial u_1}{\partial z} - \frac{\partial u_3}{\partial x}\right) = -\frac{dw}{dx}, \;\; \chi_{xy} = \frac{1}{2}\frac{\partial \omega_y}{\partial x} = -\frac{1}{2}\frac{d^2w}{dx^2} \tag{12}$$

### 2.2. Equations of Equilibrium

First we introduce the stress resultants $N_{xx}$, $M_{xx}$, and $P_{xy}$

$$N_{xx} = \int_A \sigma_{xx}\,dA, \;\; M_{xx} = \int_A z\sigma_{xx}\,dA, \;\; P_{xy} = \int_A \mathcal{M}_{xy}\,dA, \tag{13}$$

where $\mathcal{M}_{xy}$ is the couple stress induced by the difference between rates of rotations. Then using the principle minimum total potential energy, we obtain the Euler equations of equilibrium as

$$\frac{dN_{xx}}{dx} + f = 0, \tag{14}$$

$$\frac{d^2\bar{M}_{xx}}{dx^2} + \frac{d}{dx}\left(\frac{dw}{dx}N_{xx}\right) + q = 0, \tag{15}$$

where $\bar{M}_{xx} = M_{xx} + P_{xy}$ takes into account the modified couple stress effects in both the governing equations and the boundary conditions.

The duality pairs of the CBT are (the first element of each pair is the primary variable and the second element is the secondary variable)

$$(u, N_{xx}), \;\; (w, V_{\text{eff}}), \;\; \left(-\frac{dw}{dx}, \bar{M}_{xx}\right). \tag{16}$$

where $V_{\text{eff}}$ is the effective shear force

$$V_{\text{eff}} \equiv \frac{d\bar{M}_{xx}}{dx} + N_{xx}\frac{dw}{dx} \tag{17}$$

### 2.3. Governing Equations in Terms of Displacements

For an isotropic material the one-dimensional stress–strain relation

$$\sigma_{xx} = E(z)\,\varepsilon_{xx}(x,z) \tag{18}$$

We assume that the beam is graded with two material combination through the beam height according to the relation

$$E(z) = (E_1 - E_2)V_1(z) + E_2, \quad V_1(z) = \left(\frac{1}{2} + \frac{z}{h}\right)^n \tag{19}$$

where $E_1$ and $E_2$ are Young's moduli of the two materials used, and $n$ is the index that dictates the relative dominance of volume fractions $V_1(z)$ and $V_2(z) = 1 - V_1(z)$. We assume that Poisson's ratio $\nu$ is a constant for the FGM material.

The stress resultants can be expressed in terms of the displacements as

$$N_{xx} = \int_A \sigma_{xx}\,dA = A_{xx}\left[\frac{du}{dx} + \frac{1}{2}\left(\frac{dw}{dx}\right)^2\right] - B_{xx}\frac{d^2w}{dx^2} \tag{20a}$$

$$M_{xx} = \int_A z\sigma_{xx}\,dA = B_{xx}\left[\frac{du}{dx} + \frac{1}{2}\left(\frac{dw}{dx}\right)^2\right] - D_{xx}\frac{d^2w}{dx^2} \tag{20b}$$

$$P_{xy} = \int_A \mathcal{M}_{xy}\,dA = -A_{xy}\frac{d^2w}{dx^2} \tag{20c}$$

where $A_{xx}$, $B_{xx}$, $D_{xx}$, and $A_{xy}$ are the extensional, extensional-bending, bending, and in-plane shear stiffness coefficients. For beams with $E = E(x)$ (i.e., $n = 0$ or $E_1 = E_2 = E$), we have $A_{xx} = EA_0$, $B_{xx} = 0$, $D_{xx} = EI_0$, and $A_{xy} = GA_0\ell^2$. For $n \neq 0$ (i.e., FGM beams), we have

$$A_{xx} = \int_A E(z)\,dA = E_2 A_0 \frac{M+n}{1+n}$$

$$B_{xx} = \int_A E(z)z\,dA = E_2 B_0 \frac{n(M-1)}{2(1+n)(2+n)}$$

$$D_{xx} = \int_A E(z)z^2\,dA = E_2 I_0 \left[\frac{(6+3n+3n^2)M + (8n+3n^2+n^3)}{6+11n+6n^2+n^3}\right] \tag{21}$$

$$A_{xy} = \frac{\ell^2}{2(1+\nu)}\int_A E(z)\,dA = \ell^2 \frac{E_2 A_0}{2(1+\nu)}\frac{M+n}{1+n}$$

$$A_0 = bh, \quad B_0 = bh^2, \quad I_0 = \frac{bh^3}{12}, \quad M = \frac{E_1}{E_2}.$$

Figure 1a contains the variation of the non-dimensional axial stiffness ($\bar{A}_{xx} = A_{xx}/E_2 A_0$, $A_0 = bh$) and bending stiffness $\bar{D}_{xx} = D_{xx}/E_2 I_0$, $I_0 = bh^3/12$) as functions of the volume fraction index $n$ for various values of the modulus ratio $M = E_1/E_2 \geq 1$ and Figure 1b contains similar plots for the non-dimensional extensional-bending coupling stiffness $\bar{B}_{xx} = B_{xx}/E_2 B_0$, $B_0 = bh$). It is clear that both $\bar{A}_{xx}$ and $\bar{D}_{xx}$ are the maximum at $n = 0$ and decrease with increasing values of $n$. However, $\bar{B}_{xx}$ is zero at $n = 0$, increases to a maximum at $n = \sqrt{2}$, and then decreases with increasing value of $n$. Thus, beams with nonzero $B_{xx}$ will have a response that is not monotonic with respect to $n$.

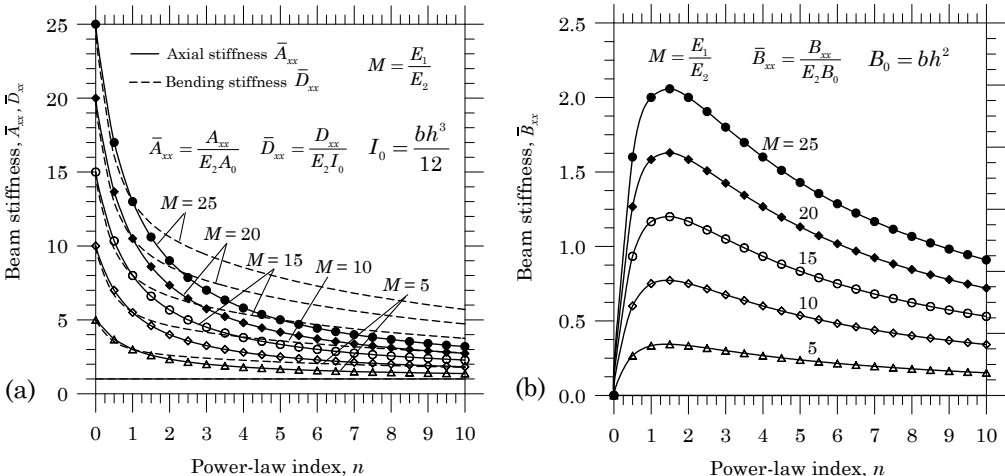

**Figure 1.** Variation of the normalized (**a**) axial stiffness coefficients $\bar{A}_{xx}(n) = A_{xx}/E_2A_0$ and $\bar{D}_{xx}(n) = D_{xx}/E_2I_0$ and (**b**) coupling stiffness coefficients $\bar{B}_{xx}(n) = B_{xx}/E_2B_0$ as functions of the power-law index, $n$, for various values of the modulus ratio, $M = E_1/E_2$.

The equations of equilibrium can be expressed in terms of the displacements $u$ and $w$ using the beam constitutive relations from Equation (21). We obtain

$$-\frac{d}{dx}\left\{A_{xx}\left[\frac{du}{dx} + \frac{1}{2}\left(\frac{dw}{dx}\right)^2\right] + B_{xx}\left(-\frac{d^2w}{dx^2}\right) - N_{xx}^T\right\} - f = 0 \tag{22}$$

$$-\frac{d^2}{dx^2}\left\{B_{xx}\left[\frac{du}{dx} + \frac{1}{2}\left(\frac{dw}{dx}\right)^2\right] + (D_{xx} + A_{xy})\left(-\frac{d^2w}{dx^2}\right) - M_{xx}^T\right\}$$

$$-\frac{d}{dx}\left\{A_{xx}\frac{dw}{dx}\left[\frac{du}{dx} + \frac{1}{2}\left(\frac{dw}{dx}\right)^2\right] - B_{xx}\frac{dw}{dx}\frac{d^2w}{dx^2}\right\} - q = 0 \tag{23}$$

*2.4. Exact Solutions*

2.4.1. General Solution

The exact solution to the linearized equations of equilibrium (i.e., static case) under distributed load $q(x)$ is given by

$$u(x) = \frac{\hat{D}_{xx}}{\hat{D}_{xx}^*}K_1x + \frac{B_{xx}}{\hat{D}_{xx}^*}\int^x\int^\xi\int^\eta q(\zeta)\,d\zeta\,d\eta\,d\xi$$

$$-\frac{B_{xx}}{\hat{D}_{xx}^*}\left(K_2\frac{x^2}{2} + K_3x + K_4\right) \tag{24}$$

$$w(x) = \left(\frac{B_{xx}}{\hat{D}_{xx}^*}K_1\right)\frac{x^2}{2} - \frac{A_{xx}}{\hat{D}_{xx}^*}\left(K_2\frac{x^3}{6} + K_3\frac{x^2}{2} + K_5x + K_6\right)$$

$$+\frac{A_{xx}}{\hat{D}_{xx}^*}\int^x\int^\xi\int^\eta\int^\zeta q(\mu)\,d\mu\,d\zeta\,d\eta\,d\xi \tag{25}$$

where $K_1$ through $K_6$ are constants of integration and $\xi$, $\eta$, $\zeta$, and $\mu$ are dummy coordinates introduced to indicate the order of integration. The six constants of integration are determined using six boundary conditions, three at each end of the beam (i.e., one element of the each of the three duality pairs at each point: $(u, N_{xx})$, $(w, V_{\text{eff}} = d\bar{M}_{xx}/dx)$, and $(dw/dx, \bar{M}_{xx})$. The stress resultants $N_{xx}$ and $\bar{M}_{xx}$ can be computed using Equations (20a) and (20b).

We can determine the constants of integration for various boundary conditions (pinned: $u = w = \bar{M}_{xx} = 0$; hinged: $N_{xx} = w = \bar{M}_{xx} = 0$; and clamped $u = w = dw/dx = 0$). In the following we present the exact solutions for beams with various boundary conditions at $x = 0$ and $x = L$, $L$ being the length of the beam.

### 2.4.2. Pinned-Hinged Beams

The exact solution for this case, with FGM and modified couple stress (MCS) effect, is ($\xi = x/L$):

$$\hat{D}_{xx}^* u(\xi) = B_{xx} \frac{q_0 L^3}{12} \left( -3\xi^2 + 2\xi^3 \right) \tag{26}$$

$$\hat{D}_{xx}^* w(\xi) = A_{xx} \frac{q_0 L^4}{24} \left( \xi - 2\xi^3 + \xi^3 \right), \tag{27}$$

$$\bar{M}_{xx} = \frac{q_0 L^2}{2} \left( \xi - \xi^2 \right), \quad N_{xx} = 0, \tag{28}$$

$$N_{xz} = \frac{dM_{xx}}{dx} = \frac{q_0 L}{2} (1 - 2\xi). \tag{29}$$

We note that the effect of $B_{xx}$ on the mechanical deflection is zero, while the bending moment is independent of $B_{xx}$.

### 2.4.3. Pinned-Pinned Beams

The solution for the pinned-pinned FGM beam is ($\xi = x/L$)

$$\hat{D}_{xx}^* u(\xi) = B_{xx} \frac{q_0 L^3}{12} \left( \xi - 3\xi^2 + 2\xi^3 \right), \tag{30}$$

$$\hat{D}_{xx}^* w(\xi) = -\frac{B_{xx}^2}{\hat{D}_{xx}} \frac{q_0 L^4}{24} \left( \xi - \xi^2 \right) + A_{xx} \frac{q_0 L^4}{24} \left( \xi - 2\xi^3 + \xi^4 \right), \tag{31}$$

$$\bar{M}_{xx} = \frac{q_0 L^2}{2} \left( \xi - \xi^2 \right), \quad N_{xx} = 0, \tag{32}$$

$$N_{xz} = \frac{dM_{xx}}{dx} = \frac{q_0 L}{2} (1 - 2\xi). \tag{33}$$

When $B_{xx} = 0$, the solutions for the pinned-hinged beams and pinned-pinned beams coincide.

### 2.4.4. Numerical Results

To present numerical results, we consider pinned-pinned functionally graded beams of length $L = 100$ in (254 cm), height $h = 1$ in (2.54 cm), and width $b = 1$ in (2.54 cm) and subjected to uniformly distributed load of intensity $q_0$ lb/in (1 lb/in = 175 N/m). The FGM beam is made of two materials with the following values of the moduli, Poisson's ratio, and shear correction coefficient:

$$E_1 = 30 \times 10^6 \text{ psi (210 GPa)}, \quad E_2 = 3 \times 10^6 \text{ psi (21 GPa)}, \quad \nu = 0.3$$

We shall investigate the parametric effects of the volume fraction index, $n$, and boundary conditions on the transverse deflections and bending moment.

Figures 2 contain plots of $u(x)$ vs. $x/L$ and $w(x)$ vs. $x/L$, while Figure 3 slope $-(dw/dx)(x)$ vs. $x/L$ and bending moment $M_{xx}(x)$ vs. $x/L$. The axial displacement $u(x)$ exists only because of the coupling coefficient $B_{xx}$ (because $f = 0$), and the way $B_{xx}$ varies with $n$ (see Figure 1b) is reflected in the variation of $u(x)$ with $n$. In particular, $u(x)$ increases with increasing values of $n$ for $n < 5$ but the magnitude of $u(x)$ decreases with increasing values of $n$. On the other hand, the deflection and slope increase their magnitudes with the increasing values of $n$ as the bending stiffness $D_{xx}$ dominates bending.

Figure 4 contains variations of the maximum displacements $u(0.25L)$ and $w(0.5L)$ with the volume fraction index $n$. It is clear from the plots that the displacement $u$ increases with $n$ for $n \leq 5$ and then decreases with the increasing values of $n$, as dictated by the variation of $B_{xx}$ with $n$ (see Figure 1b). Both $w(0.5L)$ and $-(dw/dx)(L)$ (not shown here) increase with $n$ but there are two parts, the first part exhibits rapid increase with $n$ followed by slow increase due to the interplay between $B_{xx}$ and $D_{xx}$ in the bending response.

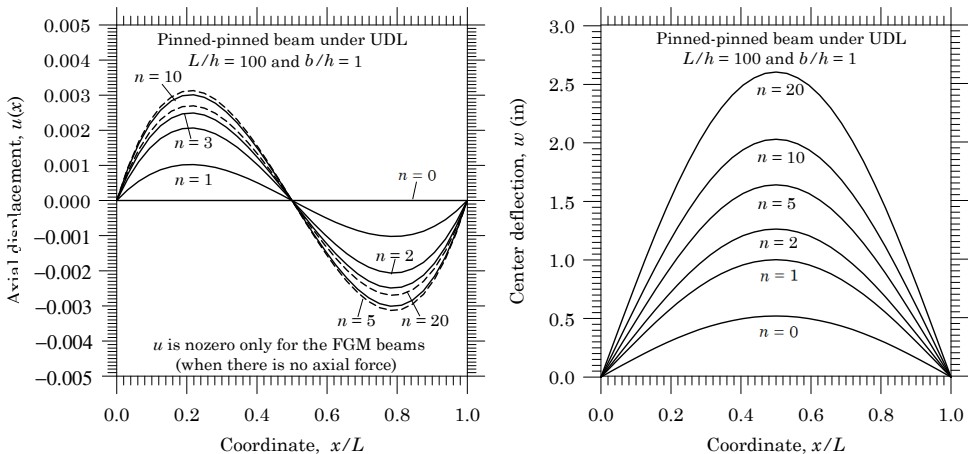

**Figure 2.** Plots of maximum displacements $u(x)$ and $w(x)$ versus $\xi = x/L$ for pinned-pinned FGM beams under uniformly distributed transverse load.

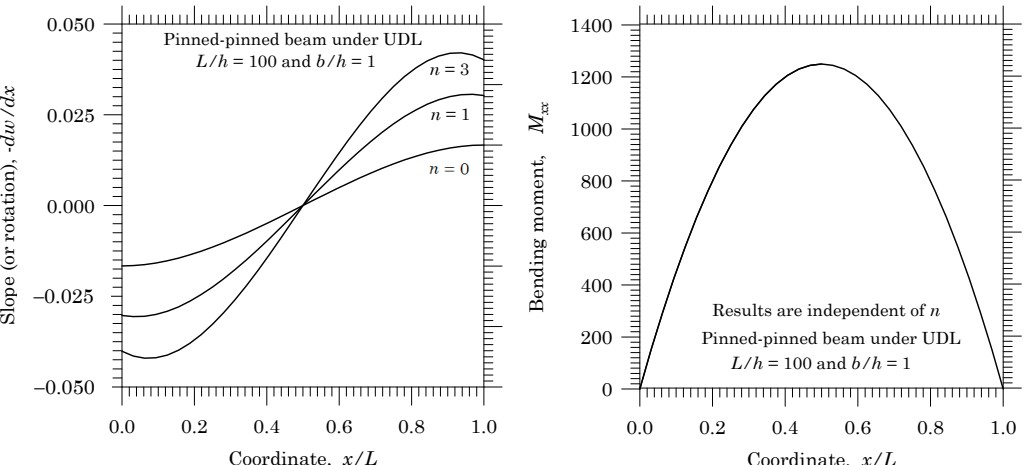

**Figure 3.** Plots of maximum slope and bending moment $-(dw/dx)(x)$ and $M_{xx}(x)$ versus $\xi = x/L$ for pinned-pinned FGM beams under uniformly distributed transverse load.

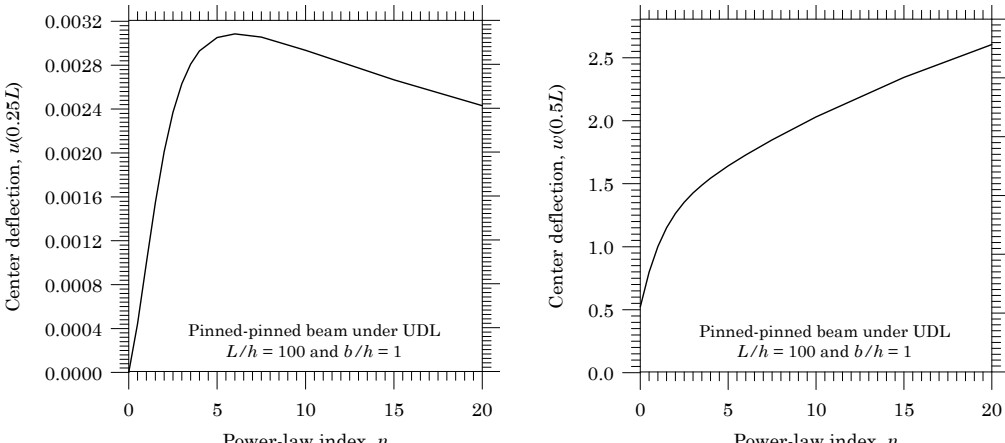

**Figure 4.** Plots of maximum displacements $u(0.25L)$ and $w(0.5L)$ versus $n$ for pinned-pinned FGM beams under uniformly distributed transverse load.

### 2.4.5. Clamped Beams

For beams clamped at both ends and subjected to uniformly distributed transverse load of intensity $q(x) = q_0$, we have

$$\hat{D}^*_{xx} u(\xi) = B_{xx} \frac{q_0 L^3}{12} \left( \xi - 3\xi^2 + 2\xi^3 \right), \tag{34}$$

$$-\hat{D}^*_{xx} \frac{dw}{dx} = A_{xx} \frac{q_0 L^3}{12} \left( -\xi + 3\xi^2 - 2\xi^3 \right), \tag{35}$$

$$\hat{D}^*_{xx} w(\xi) = A_{xx} \frac{q_0 L^4}{24} \left( \xi^2 - 2\xi^3 + \xi^4 \right), \tag{36}$$

$$M_{xx}(x) = -\frac{q_0 L^2}{12} \left( 1 - 6\xi + 6\xi^2 \right), \tag{37}$$

$$N_{xz} = \frac{dM_{xx}}{dx} = \frac{q_0 L}{2}(1 - 2\xi), \quad N_{xx} = 0. \tag{38}$$

The variation of $u(x)$ for the clamped-clamped beam is the same as that of the pinned-pinned beam. Figure 5 contains plots of $w(x)$ vs. $x/L$ and $-(dw/dx)(x)$ vs. $x/L$ while Figure 6 contains the bending moment $M_{xx}(x)$ vs. $x/L$. The data used here is the same as that used for pinned-pinned beams. The results obtained show the same trends as those discussed for the pinned-pinned beams.

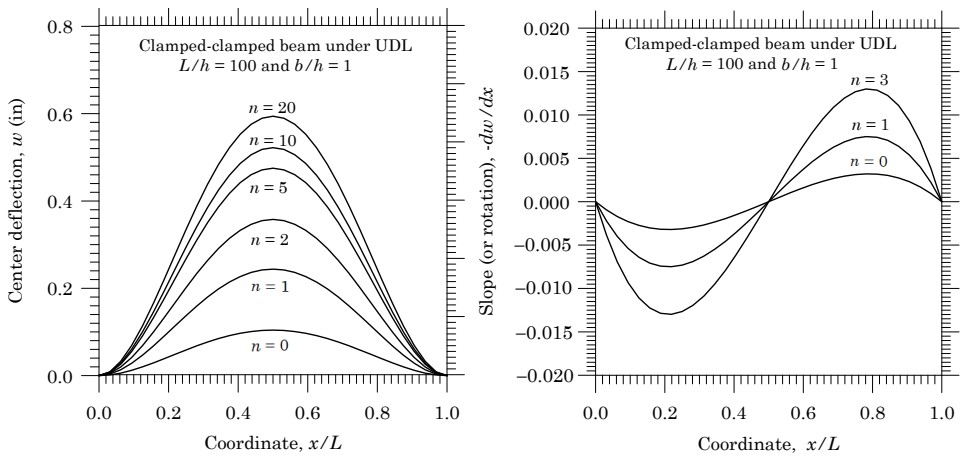

**Figure 5.** Plots of maximum displacements $w(x)$ versus $x/L$ and the maximum slope $-(dw/dx)(L)$ versus $x/L$ for clamped-clamped FGM beams under uniformly distributed transverse load.

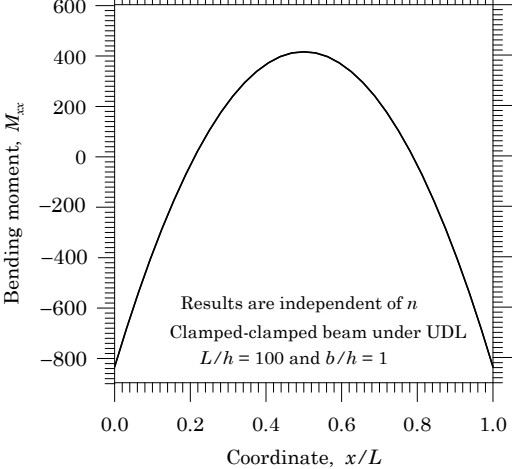

**Figure 6.** Plots of maximum bending moment $M_{xx}(x)$ versus $x/L$ for clamped-clamped FGM beams under uniformly distributed transverse load.

### 3. First-Order Theory of Beams (TBT)

#### 3.1. Preliminary Comments

In this section we consider the first-order shear deformation theory (TBT), most commonly known as the *Timoshenko beam theory*. The TBT brings the transverse shear strain $\gamma_{xz} = 2\varepsilon_{xz}$ and shear stress $\sigma_{xz}$ into the calculations. However, in the TBT the transverse shear stress through the beam thickness is only represented as a constant, whereas the elasticity equations (as discussed in mechanics of materials books) show that the variation should be quadratic. To account for the inaccuracy in predicting the transverse shear force magnitude (not the shear stress distribution itself), *shear correction factor* (SCF) has been introduced (see [2,36,37], among many others). According to Timoshenko, the shear correction factor is the ratio of the average shear strain on a section to the shear strain at the geometric centroid of the cross section. The SCF, in general, is a function of the cross-sectional shape, Poisson's ratio, material properties, boundary conditions, and so on. For rectangular sections, Timoshenko [2] proposed a SCF $K_s = \frac{5(1+\nu)}{(6+5\nu)}$, which takes the range values $(5/6) \leq K_s \leq (15/17)$ for $0 \leq \nu \leq 0.5$. Following these preliminary comments, we proceed, in somewhat parallel fashion to the developments presented for the classical beam theory (CBT), to develop the equations of equilibrium and exact solutions for the linear case.

#### 3.2. Displacements and Strains

The TBT is based on the displacement field

$$\mathbf{u}(x,z) = [u(x) + z\phi_x(x)]\hat{\mathbf{e}}_x + w(x)\hat{\mathbf{e}}_z \tag{39}$$

where $\phi_x$ denotes the rotation (independent of the slope, $\theta_x = -dw/dx$) of the cross-sectional plane about the $y$-axis. In the TBT, the normality assumption of the classical beam theory (CBT) is relaxed and a constant state of transverse shear strain (and thus constant shear stress computed from the constitutive equation) with respect to the thickness coordinate $z$ is included. As stated earlier, the TBT requires a shear correction factor to compensate for the error due to this constant shear stress assumption.

The von Kármán nonlinear strains of the TBT are

$$\varepsilon_{xx}(x,z) = \frac{du}{dx} + \frac{1}{2}\left(\frac{dw}{dx}\right)^2 + z\frac{d\phi_x}{dx} \equiv \varepsilon_{xx}^{(0)} + z\varepsilon_{xx}^{(1)} \tag{40a}$$

$$\gamma_{xz}(x) = \phi_x + \frac{dw}{dx}, \quad \varepsilon_{xx}^{(0)} = \frac{du}{dx} + \frac{1}{2}\left(\frac{dw}{dx}\right)^2, \quad \varepsilon_{xx}^{(1)} = \frac{d\phi_x}{dx} \tag{40b}$$

where $G$ the shear modulus $[G(z) = E(z)/2(1+\nu)]$ and $\nu$ is Poisson's ratio, which is assumed to be a constant.

#### 3.3. Equations of Equilibrium

The principle of minimum total potential energy for the TBT has the same form as that for CBT, except that one must add the strain energy terms associated with the transverse stress $\sigma_{xz}$. The curvature in the TBT is given by

$$\omega_y = \frac{1}{2}\left(\frac{\partial u_1}{\partial z} - \frac{\partial u_3}{\partial x}\right) = \frac{1}{2}\left(\phi_x - \frac{dw}{dx}\right), \quad \chi_{xy} = \frac{1}{2}\frac{\partial \omega_y}{\partial x} = \frac{1}{4}\left(\frac{d\phi_x}{dx} - \frac{d^2w}{dx^2}\right) \tag{41}$$

The stress resultants of the TBT are defined as

$$N_{xx} = \int_A \sigma_{xx} \, dA, \quad N_{xz} = K_s \int_A \sigma_{xz} \, dA,$$
$$M_{xx} = \int_A z\sigma_{xx} \, dA, \quad P_{xy} = \int_A \mathcal{M}_{xy} \, dA. \tag{42}$$

Here $K_s$ denotes the shear correction factor. The Euler equations of the TBT are

$$-\frac{dN_{xx}}{dx} - f = 0 \tag{43}$$

$$-\frac{dN_{xz}}{dx} - \frac{1}{2}\frac{d^2 P_{xy}}{dx^2} - \frac{d}{dx}\left(N_{xx}\frac{dw}{dx}\right) - q = 0 \tag{44}$$

$$-\frac{dM_{xx}}{dx} - \frac{1}{2}\frac{dP_{xy}}{dx} + N_{xz} = 0 \tag{45}$$

The three duality pairs for the TBT are

$$(u, N_{xx}), \quad (w, V_{\text{eff}}), \quad (\phi_x, \bar{M}_{xx}). \tag{46}$$

where the effective shear force and bending moments are

$$V_{\text{eff}} \equiv N_{xz} + N_{xx}\frac{dw}{dx} + \frac{1}{2}\frac{dP_{xy}}{dx} \tag{47a}$$

$$\bar{M}_{xx} = M_{xx} + \frac{1}{2}P_{xy} \tag{47b}$$

We note that the effective shear force in the TBT has the modified couple stress term.

### 3.4. Governing Equations in Terms of Displacements
Beam Constitutive Equations

The stress resultants $(N_{xx}, M_{xx}, N_{xz}, P_{xy})$ in terms of the strains are

$$N_{xx} = \int_A \sigma_{xx} \, dA = A_{xx}\left[\frac{du}{dx} + \frac{1}{2}\left(\frac{dw}{dx}\right)^2\right] + B_{xx}\frac{d\phi_x}{dx} \tag{48a}$$

$$M_{xx} = \int_A z\sigma_{xx} \, dA = B_{xx}\left[\frac{du}{dx} + \frac{1}{2}\left(\frac{dw}{dx}\right)^2\right] + D_{xx}\frac{d\phi_x}{dx} \tag{48b}$$

$$P_{xy} = \int_A \mathcal{M}_{xy} \, dA = \frac{1}{2}A_{xy}\left(\frac{d\phi_x}{dx} - \frac{d^2 w}{dx^2}\right) \tag{48c}$$

$$N_{xz} = K_s \int_A \sigma_{xz} \, dA = S_{xz}\left(\phi_x + \frac{dw}{dx}\right), \tag{48d}$$

where $A_{xx}$, $B_{xx}$, $D_{xx}$, and $A_{xy}$ are as defined in Equation (21), and $K_s$ denotes the shear correction factor and $S_{xz}$ is the shear stiffness

$$S_{xz} = \frac{K_s}{2(1+\nu)}\int_A E(z) \, dA \tag{49}$$

The equations of equilibrium in Equations (43)–(45) now can be expressed in terms of the displacements $u$, $w$, and $\phi_x$ with the help of the beam constitutive relations in Equations (48a)–(48c) as

$$-\frac{d}{dx}\left\{A_{xx}\left[\frac{du}{dx}+\frac{1}{2}\left(\frac{dw}{dx}\right)^2\right]+B_{xx}\frac{d\phi_x}{dx}\right\}=f, \tag{50}$$

$$-\frac{d}{dx}\left[S_{xz}\left(\phi_x+\frac{dw}{dx}\right)\right]-\frac{1}{4}\frac{d^2}{dx^2}\left[A_{xy}\left(\frac{d\phi_x}{dx}-\frac{d^2w}{dx^2}\right)\right]$$
$$-\frac{d}{dx}\left\{A_{xx}\frac{dw}{dx}\left[\frac{du}{dx}+\frac{1}{2}\left(\frac{dw}{dx}\right)^2\right]+B_{xx}\frac{dw}{dx}\left(\frac{d\phi_x}{dx}\right)\right\}-q=0 \tag{51}$$

$$-\frac{d}{dx}\left\{B_{xx}\left[\frac{du}{dx}+\frac{1}{2}\left(\frac{dw}{dx}\right)^2\right]+D_{xx}\frac{d\phi_x}{dx}\right\}+S_{xz}\left(\phi_x+\frac{dw}{dx}\right)$$
$$-\frac{1}{4}\frac{d}{dx}\left[A_{xy}\left(\frac{d\phi_x}{dx}-\frac{d^2w}{dx^2}\right)\right]=0 \tag{52}$$

*3.5. Exact Solutions*

3.5.1. General Solution

In this section we present exact solutions to the *linear* equations of equilibrium of FGM beams without the modified couple stress effect. By setting $f=0$ in Equations (50)–(52), we obtain

$$-\frac{d}{dx}\left[A_{xx}\left(\frac{du}{dx}\right)+B_{xx}\frac{d\phi_x}{dx}\right]=f, \tag{53}$$

$$-\frac{d}{dx}\left[S_{xz}\left(\phi_x+\frac{dw}{dx}\right)\right]-q=0, \tag{54}$$

$$-\frac{d}{dx}\left(B_{xx}\frac{du}{dx}+D_{xx}\frac{d\phi_x}{dx}\right)+S_{xz}\left(\phi_x+\frac{dw}{dx}\right)=0. \tag{55}$$

Again, we further assume that the beam stiffness coefficients are all constant and $f=0$.

Equations (53)–(55), when expressed in terms of the stress resultants $N_{xx}$, $N_{xz}$, and $M_{xx}$ (see Equations (43)–(45) with $P_{xy}=0$) take the following form:

$$\frac{dN_{xx}}{dx}=0,\quad -\frac{dN_{xz}}{dx}-q=0,\quad -\frac{dM_{xx}}{dx}+N_{xz}=0 \tag{56}$$

Substituting for $N_{xz}$ from the third equation into the second equation, we obtain

$$\frac{dN_{xx}}{dx}=0,\quad -\frac{d^2M_{xx}}{dx^2}-q=0. \tag{57}$$

Integrating the above equations,

$$N_{xx}=c_1,\quad \frac{dM_{xx}}{dx}=-\int^x q(\xi)d\xi+c_2 \tag{58}$$

Integrating the second equation, we obtain

$$M_{xx}=-\int^x\int^\xi q(\eta)\,d\eta d\xi+c_2x+c_3\equiv F(x), \tag{59}$$

Here $c_i$ $(i=1,2,3)$ denote the constants of integration.

The left sides of Equations (58) and (59) can be expressed in terms of the displacements $(u, \phi_x)$ using Equations (48a) and (48b); we have

$$
\begin{bmatrix} A_{xx} & B_{xx} \\ B_{xx} & D_{xx} \end{bmatrix} \left\{ \begin{array}{c} \frac{du}{dx} \\ \frac{d\phi_x}{dx} \end{array} \right\} = \left\{ \begin{array}{c} c_1 + N_{xx}^T \\ F(x) + M_{xx}^T \end{array} \right\} \tag{60}
$$

Solving for $du/dx$ and $d\phi_x/dx$, we obtain

$$
\frac{du}{dx} = \frac{1}{D_{xx}^*} [D_{xx} c_1 - B_{xx} F(x)] \tag{61}
$$

$$
\frac{d\phi_x}{dx} = \frac{1}{D_{xx}^*} [-B_{xx} c_1 + A_{xx} F(x)] \tag{62}
$$

where

$$
D_{xx}^* = A_{xx} D_{xx} - B_{xx}^2, \quad F(x) = -\int^x \int^\xi q(\eta) \, d\eta \, d\xi + c_2 x + c_3 \tag{63}
$$

Integrating Equations (61) and (62), we obtain

$$
u(x) = \frac{D_{xx}}{D_{xx}^*} c_1 x + \frac{B_{xx}}{D_{xx}^*} \int^x \int^\xi \int^\eta q(\zeta) \, d\zeta \, d\eta \, d\xi
$$
$$
- \frac{B_{xx}}{D_{xx}^*} \left( c_2 \frac{x^2}{2} + c_3 x + c_4 \right), \tag{64}
$$

$$
\phi_x(x) = -\frac{B_{xx}}{D_{xx}^*} c_1 x - \frac{A_{xx}}{D_{xx}^*} \int^x \int^\xi \int^\eta q(\zeta) \, d\zeta \, d\eta \, d\xi
$$
$$
+ \frac{A_{xx}}{D_{xx}^*} \left( c_2 \frac{x^2}{2} + c_3 x + c_5 \right). \tag{65}
$$

From Equations (56) and (58), we arrive at

$$
N_{xz} = \frac{dM_{xx}}{dx} = -\int^x q(\xi) \, d\xi + c_2 \tag{66}
$$

and using Equation (48d) we obtain

$$
\frac{dw}{dx} = \frac{1}{S_{xz}} \left( -\int^x q(\xi) \, d\xi + c_2 \right) - \phi_x
$$
$$
= \frac{1}{S_{xz}} \left( -\int^x q(\xi) \, d\xi + c_2 \right) + \frac{B_{xx}}{D_{xx}^*} c_1 x
$$
$$
- \frac{A_{xx}}{D_{xx}^*} \left( -\int^x \int^\xi \int^\eta q(\zeta) \, d\zeta \, d\eta \, d\xi + c_2 \frac{x^2}{2} + c_3 x + c_5 \right) \tag{67}
$$

or

$$
w(x) = \frac{1}{S_{xz}} \left( -\int^x \int^\xi q(\eta) \, d\eta \, d\xi + c_2 x \right) + \frac{B_{xx}}{D_{xx}^*} c_1 \frac{x^2}{2}
$$
$$
- \frac{A_{xx}}{D_{xx}^*} \left( -\int^x \int^\xi \int^\eta \int^\zeta q(\mu) \, d\mu \, d\zeta \, d\eta \, d\xi + c_2 \frac{x^3}{6} + c_3 \frac{x^2}{2} + c_5 x + c_6 \right) \tag{68}
$$

The six constants of integration are determined using six boundary conditions, three at each end of the beam. One (and only one) element of the each of three duality pairs at each boundary point must be known (see Equation (43)): $(u, N_{xx})$, $(w, N_{xz})$, and $(\phi_x, M_{xx})$. We note that, in TBT, $\phi_x$ has replaced $-dw/dx$ as the primary variable, and it is dual to the bending moment $M_{xx}$. *One should not specify $dw/dx$ in place of $\phi_x$ in the TBT.* The stress resultants $(N_{xx}, M_{xx}, P_{xy}, N_{xz})$ can be computed with the help of Equations (48a)–(48d).

3.5.2. Pinned-Pinned Beams

The exact solution of a beam pinned at both ends ($u(0) = 0$, $w(0) = 0$, $M_{xx}(0) = 0$, $u(L) = 0$, $w(L) = 0$, and $M_{xx}(L) = 0$) is

$$D^*_{xx} u(\xi) = B_{xx} \frac{q_0 L^3}{12} \left( \xi - 3\xi^2 + 2\xi^3 \right),  \tag{69}$$

$$D^*_{xx} \phi_x(\xi) = \frac{B^2_{xx}}{D_{xx}} \frac{q_0 L^3}{24} (1 - 2\xi)$$
$$- A_{xx} \frac{q_0 L^3}{24} \left( 1 - 6\xi + 4\xi^3 \right),  \tag{70}$$

$$D^*_{xx} w(\xi) = - \frac{B^2_{xx}}{D_{xx}} \frac{q_0 L^4}{24} \left( \xi - \xi^2 \right)$$
$$+ A_{xx} \frac{q_0 L^4}{24} \left( \xi - 2\xi^3 + \xi^4 \right) + \frac{D^*_{xx}}{S_{xz}} \frac{q_0 L^2}{2} \left( \xi - \xi^2 \right),  \tag{71}$$

$$\bar{M}_{xx} = \frac{q_0 L^2}{2} \left( \xi - \xi^2 \right), \quad N_{xz} = \frac{q_0 L}{2} (1 - 2\xi), \quad N_{xx} = 0.  \tag{72}$$

It is clear that all functions except the transverse deflection are the same as those predicted by the classical beam theory (CBT). The transverse deflection has an additional positive term that *adds* to the value predicted by the CBT. Thus, the first-order shear deformation theory (TBT) deflection $w$ is larger than those predicted by the CBT (i.e., the CBT underpredicts $w$).

The numerical results obtained by the TBT are either the same or very close to those obtained using the CBT. Because of the fact that the beam considered is a thin beam with length-to-height ratio of $L/h = 100$, the effect of the shear deformation is not seen. Table 1 shows the numerical results obtained with the two theories for the data:

$$E_1 = 30 \times 10^6 \text{ psi (210 GPa)}, \quad E_2 = 3 \times 10^6 \text{ psi (21 GPa)}, \quad \nu = 0.3, \quad K = \frac{5}{6}$$

Transverse deflections obtained with the CBT and TBT for three different length-to-height ratios, $L/h = 50$, $L/h = 20$, and $L/h = 10$ are presented in Table 2. It is clear that when the beam is moderately thick ($L/h = 20$) to very thick ($L/h = 10$), the CBT under predicts the deflections, although the difference between the two solutions may not be significant.

**Table 1.** Numerical results obtained with the classical (CBT) and the shear deformation (TBT) beam theories for the displacements $\bar{u} = u(0.25L) \times 10^2$ and $w(0.5L)$ and slopes $\theta_x(L) = -(dw/dx)(L)$ and $\phi_x(L)$ of a *pinned–pinned* FGM beams under a uniformly distributed load (all results are normalized with the load).

| $n$ | $\bar{u}$-CBT | $\bar{u}$-TBT | $w$-CBT | $w$-TBT | $\theta_x$ | $\phi_x$ |
|------|---------|---------|--------|--------|---------|---------|
| 0.0  | 0.00000 | 0.00000 | 0.5208 | 0.5210 | 0.01657 | 0.01657 |
| 1.0  | 0.09973 | 0.09973 | 1.0014 | 1.0016 | 0.03062 | 0.03062 |
| 2.0  | 0.20118 | 0.20118 | 1.2635 | 1.2638 | 0.03722 | 0.03722 |
| 3.0  | 0.26301 | 0.26301 | 1.4261 | 1.4265 | 0.04148 | 0.04148 |
| 4.0  | 0.29297 | 0.29297 | 1.5440 | 1.5445 | 0.04495 | 0.04495 |
| 5.0  | 0.30523 | 0.30523 | 1.6415 | 1.6420 | 0.04806 | 0.04806 |
| 6.0  | 0.30850 | 0.30850 | 1.7286 | 1.7292 | 0.05097 | 0.05097 |
| 10.0 | 0.29367 | 0.29367 | 2.0305 | 2.0312 | 0.06131 | 0.06131 |
| 20.0 | 0.24302 | 0.24302 | 2.6047 | 2.6056 | 0.08080 | 0.08080 |

**Table 2.** Numerical results obtained with the classical (CBT) and the Timoshenko (TBT) beam theories for the transverse deflections of a *pinned–pinned* FGM beams under a uniformly distributed load (the results are normalized with the load, $q_0$); $\bar{w} = w(0.5L)\,10$ and $\hat{w} = w(0.5L) \times 10^2$.

| | $L/h = 50$ | | $L/h = 20$ | | $L/h = 10$ | |
|---|---|---|---|---|---|---|
| $n$ | $w$-CBT | $w$-TBT | $\bar{w}$-CBT | $\bar{w}$-TBT | $\hat{w}$-CBT | $\hat{w}$-TBT |
| 0.0 | 0.06510 | 0.06517 | 0.04167 | 0.04193 | 0.05208 | 0.05338 |
| 1.0 | 0.12517 | 0.12529 | 0.08011 | 0.08058 | 0.10014 | 0.10250 |
| 2.0 | 0.15793 | 0.15809 | 0.10108 | 0.10173 | 0.12635 | 0.12960 |
| 3.0 | 0.17827 | 0.17847 | 0.11409 | 0.11409 | 0.14261 | 0.14661 |
| 4.0 | 0.19300 | 0.19324 | 0.12352 | 0.12445 | 0.15440 | 0.15905 |
| 5.0 | 0.20518 | 0.20544 | 0.13132 | 0.13236 | 0.16415 | 0.16935 |
| 6.0 | 0.21608 | 0.21636 | 0.13829 | 0.13943 | 0.17286 | 0.17855 |
| 10.0 | 0.25382 | 0.25417 | 0.16244 | 0.16387 | 0.20305 | 0.21020 |
| 20.0 | 0.32559 | 0.32605 | 0.20838 | 0.21020 | 0.26047 | 0.26957 |

## 4. The Third-Order Beam Theory

### 4.1. Preliminary Comments

From the discussions presented in the previous sections, it is clear that the transverse shear stress distribution through the beam height, computed using the stress–strain relation $\sigma_{xz} = 2G\varepsilon_{xz}$, is either zero (in CBT) or constant (in TBT), although the actual variation of $\sigma_{xz}(x, z)$ with $z$, determined using the 3-D equations of equilibrium of linearized elasticity, is cubic. Therefore, it is necessary to have the displacement field (especially $u_1$) to be a cubic function of $z$. In this section, a third-order beam theory which accounts for the von Kármán geometric nonlinearity, through thickness variation of the material and modified couple stress effect, is presented. The theory presented herein accounts for the vanishing of transverse shear stress on the bottom and top surfaces of the beam (see [4,8,38,39]).

### 4.2. Kinematics

The displacement field of the Reddy–Bickford third-order beam theory (RBT) is

$$\mathbf{u}(x, z) = \left[ u(x) + z\phi_x(x) - \alpha\, z^3 \left( \phi_x + \frac{dw}{dx} \right) \right] \hat{\mathbf{e}}_x + w(x)\hat{\mathbf{e}}_z, \tag{73}$$

where $\alpha = 4/3h^2$. The nonzero strain and curvature components are

$$\varepsilon_{xx} \varepsilon_{xx}^{(0)} + z\varepsilon_{xx}^{(1)} + z^3\varepsilon_{xx}^{(3)}, \quad \gamma_{xz} = \gamma_{xz}^{(0)} + z^2\gamma_{xz}^{(2)}, \tag{74a}$$

$$\chi_{xy} = \chi_{xy}^{(0)} + z^2\chi_{xy}^{(2)}, \qquad \chi_{yz} = z\,\chi_{yz}^{(1)} \tag{74b}$$

where (omitting the higher-order terms in the thickness strain)

$$\varepsilon_{xx}^{(0)} = \frac{du}{dx} + \frac{1}{2}\left(\frac{dw}{dx}\right)^2, \quad \varepsilon_{xx}^{(1)} = \frac{d\phi_x}{dx}, \quad \varepsilon_{xx}^{(3)} = -\alpha\left(\frac{d\phi_x}{dx} + \frac{d^2w}{dx^2}\right),$$

$$\gamma_{xz}^{(0)} = \phi_x + \frac{dw}{dx}, \quad \gamma_{xz}^{(2)} = -\beta\left(\phi_x + \frac{dw}{dx}\right), \quad \beta = 3\alpha = \frac{4}{h^2} \tag{74c}$$

$$\chi_{xy}^{(0)} = \frac{1}{4}\left(\frac{d\phi_x}{dx} - \frac{d^2w}{dx^2}\right), \quad \chi_{xy}^{(2)} = -\frac{1}{4}\beta\left(\frac{d\phi_x}{dx} + \frac{d^2w}{dx^2}\right),$$

$$\chi_{yz}^{(1)} = -\frac{1}{2}\beta\left(\phi_x + \frac{dw}{dx}\right)$$

### 4.3. Equations of Equilibrium

The equations of equilibrium of the RBT are derived using the principle of minimum total potential energy. We introduce the following stress resultants:

$$(N_{xx}, M_{xx}, P_{xx}) = \int_A (1, z, z^3) \sigma_{xx} \, dA, \quad (N_{xz}, P_{xz}) = \int_A (1, z^2) \sigma_{xz} \, dA, \tag{75a}$$

$$(P_{xy}, R_{xy}) = \int_A (1, z^2) \mathcal{M}_{xy} \, dA, \quad Q_{yz} = \int_A z \mathcal{M}_{yz} \, dA, \tag{75b}$$

$$\bar{M}_{xx} = M_{xx} - \alpha \, P_{xx}, \quad \bar{N}_{xz} = N_{xz} - \beta P_{xz}, \quad \beta = 3\alpha = \frac{4}{h^2}. \tag{75c}$$

The equations of equilibrium of the RBT are:

$$-\frac{dN_{xx}}{dx} = f \tag{76}$$

$$-\frac{d}{dx}\left(N_{xx}\frac{dw}{dx}\right) - \frac{d\bar{N}_{xz}}{dx} - \alpha \frac{d^2 P_{xx}}{dx^2}$$

$$-\tfrac{1}{2}\frac{d^2 P_{xy}}{dx^2} - \tfrac{1}{2}\beta \frac{d^2 R_{xy}}{dx^2} + \beta \frac{dQ_{yz}}{dx} = q \tag{77}$$

$$-\frac{d\bar{M}_{xx}}{dx} + \bar{N}_{xz} - \tfrac{1}{2}\frac{dP_{xy}}{dx} + \tfrac{1}{2}\beta\frac{dR_{xy}}{dx} - \beta \, Q_{yz} = 0 \tag{78}$$

The duality pairs (the first element of the pair denotes a generalized displacement while the second element denotes a generalized force):

$$(u, N_{xx}), \quad (w, V_{\text{eff}}), \quad \left(-\frac{dw}{dx}, M_{\text{eff}}\right), \quad (\phi_x, \tilde{M}_{xx}), \tag{79}$$

where

$$V_{\text{eff}} = \bar{N}_{xz} + \alpha \frac{dP_{xx}}{dx} + N_{xx}\frac{dw}{dx} + \tfrac{1}{2}\frac{dP_{xy}}{dx} + \tfrac{1}{2}\beta\frac{dR_{xy}}{dx} - \beta \, Q_{yz}, \tag{80}$$

$$M_{\text{eff}} = \tfrac{1}{2} P_{xy} + \tfrac{1}{2} \beta R_{xy}, \quad \tilde{M}_{xx} = \bar{M}_{xx} + \tfrac{1}{2}P_{xy} - \beta\tfrac{1}{2}R_{xy}. \tag{81}$$

Thus, there are four boundary conditions at each boundary point. Requiring $\partial w/\partial x$ as well as $\phi_x$ to vanish at a support necessarily implies that the shear force, when shear stress is computed using the constitutive relation $\sigma_{xz} = G\gamma_{xz}$, is zero. However, the effective shear force $V_{\text{eff}}$ is not zero.

One of the challenges of higher-order theories is the ability to specify boundary conditions that involve higher-order stress resultants. In most cases, one does not know the known values of the higher-order stress resultants. Therefore, whenever the lower-order stress resultant is specified, we assume that the corresponding higher-order stress resultant is known to be zero. For example, if $M_{xx}$ is specified at a point, we assume that $\bar{M}_{xx} = M_{xx}$ (implying that $P_{xx} = 0$ there).

### 4.4. Beam Constitutive Relations

In the RBT, as in in the case of CBT and TBT, we invoked the inextensibility of the transverse normal lines, which amounts to setting $\varepsilon_{zz} = 0$. Therefore, we can use one-dimensional constitutive relations. In particular, the one-dimensional constitutive relations are

$$\sigma_{xx} = E \, \varepsilon_{xx}, \qquad \sigma_{xz} = G \, \gamma_{xz}, \tag{82}$$

$$\mathcal{M}_{xy} = 2\ell^2 G \, \chi_{xy}, \quad \mathcal{M}_{yz} = 2\ell^2 G \, \chi_{yz}. \tag{83}$$

Substituting the constitutive relations from Equations (82) and (83) into the definition of the stress resultants in Equations (75a)–(75c), we obtain

$$
\begin{Bmatrix} N_{xx} \\ M_{xx} \\ P_{xx} \end{Bmatrix} = \begin{bmatrix} A_{xx} & B_{xx} & E_{xx} \\ B_{xx} & D_{xx} & F_{xx} \\ E_{xx} & F_{xx} & H_{xx} \end{bmatrix} \begin{Bmatrix} \varepsilon_{xx}^{(0)} \\ \varepsilon_{xx}^{(1)} \\ \varepsilon_{xx}^{(3)} \end{Bmatrix},
\tag{84a}
$$

$$
\begin{Bmatrix} N_{xz} \\ P_{xz} \\ P_{xy} \\ R_{xy} \\ Q_{yz} \end{Bmatrix} = \begin{bmatrix} A_{xz} & D_{xz} & 0 & 0 & 0 \\ D_{xz} & F_{xz} & 0 & 0 & 0 \\ 0 & 0 & A_{xy} & D_{xy} & 0 \\ 0 & 0 & D_{xy} & F_{xy} & 0 \\ 0 & 0 & 0 & 0 & D_{xy} \end{bmatrix} \begin{Bmatrix} \gamma_{xz}^{(0)} \\ \gamma_{xz}^{(2)} \\ 2\chi_{xy}^{(0)} \\ 2\chi_{xy}^{(2)} \\ 2\chi_{yz}^{(1)} \end{Bmatrix}.
\tag{84b}
$$

where (see Equation (21))

$$
(A_{xx}, B_{xx}, D_{xx}, E_{xx}, F_{xx}, H_{xx}) = \int_A (1, z, z^2, z^3, z^4, z^6) E(z)\, dA,
\tag{85a}
$$

$$
(A_{xz}, D_{xz}, F_{xz}) = \frac{1}{2(1+\nu)} \int_A (1, z^2, z^4) E(z)\, dA,
\tag{85b}
$$

$$
(A_{xy}, D_{xy}, F_{xy}) = \frac{\ell^2}{2(1+\nu)} \int_A (1, z^2, z^4) E(z)\, dA,
\tag{85c}
$$

### 4.5. Beam Stiffness Coefficients for FGM Beams

For the FGM beams, the integrals in Equations (85a)–(85c) can be evaluated as:

$$
\begin{aligned}
&A_{xx} = E_2 bh \frac{M+n}{1+n}, \quad B_{xx} = E_2 \frac{bh^2}{2} \frac{n(M-1)}{(1+n)(2+n)}, \\
&D_{xx} = E_2 \frac{bh^3}{12} \left[ \frac{(6+3n+3n^2)M + (8n+3n^2+n^3)}{(1+n)(2+n)(3+n)} \right], \\
&E_{xx} = E_2 \frac{bh^4}{8}(M-1) \left[ \frac{n(8+3n+n^2)}{(1+n)(2+n)(3+n)(4+n)} \right], \\
&F_{xx} = E_2 \frac{bh^5}{80} f(n), \quad H_{xx} = E_2 \frac{bh^7}{448} g(n), \\
&A_{xy} = E_2 \ell^2 \frac{bh}{2(1+\nu)} \frac{M+n}{1+n}, \quad F_{xy} = \frac{E_2 \ell^2 bh^5}{120(1+\nu)} f(n), \\
&D_{xy} = E_2 \ell^2 \frac{bh^3}{24(1+\nu)} \left[ \frac{(6+3n+3n^2)M + (8n+3n^2+n^3)}{(1+n)(2+n)(3+n)} \right],
\end{aligned}
\tag{86a}
$$

where

$$
\begin{aligned}
&f(n) = \frac{f_1 M + n f_2}{f_3}, \quad g(n) = \frac{g_1 M + g_2}{g_3}, \quad M = \frac{E_1}{E_2}, \\
&f_1 = (24 + 18n + 23n^2 + 6n^3 + n^4), \\
&f_2 = (184 + 110n + 55n^2 + 10n^3 + n^4), \\
&f_3 = (1+n)(2+n)(3+n)(4+n)(5+n), \\
&g_1 = (720 + 660n + 964n^2 + 405n^3 + 115n^4 + 15n^5 + n^6), \\
&g_2 = (720 + 1764n + 1624n^2 + 735n^3 + 175n^4 + 21n^5 + n^6), \\
&g_3 = (1+n)(2+n)(3+n)(4+n)(5+n)(6+n)(7+n).
\end{aligned}
\tag{86b}
$$

If the higher-order terms are neglected in the governing equations of motion but not in the constitutive relations, we obtain the third-order theory developed by Levinson [38].

If one neglects the higher-order terms selectively in the constitutive relations, one obtains the so-called simplified Reddy–Bickford beam theory. These ideas will be discussed shortly.

### 4.6. Equilibrium Equations in Terms of the Displacements

With the help of Equations (84a) and (84b), the equations of equilibrium, Equations (76)–(78), can be expressed in terms of the generalized displacements $(u, w, \phi_x)$. We obtain

$$-\frac{d}{dx}\left\{ A_{xx}\left[\frac{du}{dx} + \frac{1}{2}\left(\frac{dw}{dx}\right)^2\right] + \bar{B}_{xx}\frac{d\phi_x}{dx} - \alpha\,E_{xx}\frac{d^2w}{dx^2}\right\} = f, \tag{87}$$

$$-\frac{d}{dx}\left[\frac{dw}{dx}\left\{A_{xx}\left[\frac{du}{dx} + \frac{1}{2}\left(\frac{dw}{dx}\right)^2\right] + \bar{B}_{xx}\frac{d\phi_x}{dx} - \alpha\,E_{xx}\frac{d^2w}{dx^2}\right\}\right]$$
$$-\frac{1}{4}\frac{d^2}{dx^2}\left[A_{xy}\left(\frac{d\phi_x}{dx} - \frac{d^2w}{dx^2}\right) - \beta D_{xy}\left(\frac{d\phi_x}{dx} + \frac{d^2w}{dx^2}\right)\right]$$
$$-\frac{1}{4}\beta\frac{d^2}{dx^2}\left[D_{xy}\left(\frac{d\phi_x}{dx} - \frac{d^2w}{dx^2}\right) - \beta F_{xy}\left(\frac{d\phi_x}{dx} + \frac{d^2w}{dx^2}\right)\right]$$
$$-\alpha\frac{d^2}{dx^2}\left\{E_{xx}\left[\frac{du}{dx} + \frac{1}{2}\left(\frac{dw}{dx}\right)^2\right] + \bar{F}_{xx}\frac{d\phi_x}{dx} - \alpha\,H_{xx}\frac{d^2w}{dx^2}\right\}$$
$$-\hat{A}_{xz}\frac{d}{dx}\left(\phi_x + \frac{dw}{dx}\right) - \beta^2 D_{xy}\frac{d}{dx}\left(\phi_x + \frac{dw}{dx}\right) = q, \tag{88}$$

$$-\frac{d}{dx}\left\{\bar{B}_{xx}\left[\frac{du}{dx} + \frac{1}{2}\left(\frac{dw}{dx}\right)^2\right] + \hat{D}_{xx}\frac{d\phi_x}{dx} - \alpha\,\bar{F}_{xx}\frac{d^2w}{dx^2}\right\}$$
$$-\frac{1}{4}\frac{d}{dx}\left[A_{xy}\left(\frac{d\phi_x}{dx} - \frac{d^2w}{dx^2}\right) - \beta D_{xy}\left(\frac{d\phi_x}{dx} + \frac{d^2w}{dx^2}\right)\right]$$
$$+\frac{1}{4}\beta\frac{d}{x}\left[D_{xy}\left(\frac{d\phi_x}{dx} - \frac{d^2w}{dx^2}\right) - \beta F_{xy}\left(\frac{d\phi_x}{dx} + \frac{d^2w}{dx^2}\right)\right]$$
$$+\hat{A}_{xz}\left(\phi_x + \frac{dw}{dx}\right) + \beta^2 D_{xy}\left(\phi_x + \frac{dw}{dx}\right) = 0, \tag{89}$$

where

$$\bar{B}_{xx} = B_{xx} - \alpha\,E_{xx}, \quad \bar{D}_{xx} = D_{xx} - \alpha\,F_{xx}, \quad \bar{F}_{xx} = F_{xx} - \alpha\,H_{xx},$$
$$\hat{D}_{xx} = \bar{D}_{xx} - \alpha\,\bar{F}_{xx}, \quad \bar{A}_{xz} = A_{xz} - \beta\,D_{xz}, \quad \bar{D}_{xz} = D_{xz} - \beta\,F_{xz}, \tag{90}$$
$$\hat{A}_{xz} = \bar{A}_{xz} - \beta\,\bar{D}_{xz}, \quad \alpha = \frac{4}{3h^2}, \quad \beta = \frac{4}{h^2} = 3\alpha.$$

### 4.7. Exact Solutions for Bending

In this section we present exact solutions to the *linear* equations of equilibrium of the RBT for FGM beams without the effect of the modified couple stress. First, the equations of equilibrium in terms of the stress resultants can be obtained from Equations (87)–(89) by omitting the nonlinear terms and time-depedent terms:

$$-\frac{dN_{xx}}{dx} = 0 \tag{91}$$

$$-\frac{d\bar{N}_{xz}}{dx} - \alpha\frac{d^2P_{xx}}{dx^2} = q \tag{92}$$

$$-\frac{d\bar{M}_{xx}}{dx} + \bar{N}_{xz} = 0 \tag{93}$$

Integrating the equations with respect to $x$, we obtain

$$N_{xx} = K_1 \tag{94}$$

$$\bar{N}_{xz} + \alpha \frac{dP_{xx}}{dx} = -\int^x q(\xi)\,d\xi + K_2 \tag{95}$$

$$M_{xx} = -\int^x \int^\xi q(\eta)\,d\eta\,d\xi + K_2 x + K_3 \equiv F(x). \tag{96}$$

where $K_1$ and $K_2$ are constants of integration.

Expressing $N_{xx}$ and $M_{xx}$ in Equations (94) and (96) in terms of the generalized displacements, and solving for $du/dx$ and $d\phi_x/dx$ in terms of $d^2w/dx^2$, we obtain

$$\frac{du}{dx} = \frac{P_1}{\bar{D}_{xx}^*} + \frac{\bar{D}_{xx}}{\bar{D}_{xx}^*} K_1 - \frac{\bar{B}_{xx}}{\bar{D}_{xx}^*} F(x) + \frac{J_1}{\bar{D}_{xx}^*} \frac{d^2w}{dx^2}, \tag{97a}$$

$$\frac{d\phi_x}{dx} = \frac{P_2}{\bar{D}_{xx}^*} - \frac{B_{xx}}{\bar{D}_{xx}^*} K_1 + \frac{A_{xx}}{\bar{D}_{xx}^*} F(x) + \frac{J_2}{\bar{D}_{xx}^*} \frac{d^2w}{dx^2}, \tag{97b}$$

where (see Equations (87) and (88))

$$\begin{aligned} P_1 &= \bar{D}_{xx} M_T^{(0)} - \bar{B}_{xx} M_T^{(1)}, \quad P_2 = A_{xx} M_T^{(1)} - B_{xx} M_T^{(0)}, \\ J_1 &= \alpha(\bar{D}_{xx} E_{xx} - \bar{B}_{xx} F_{xx}), \quad J_2 = \alpha(A_{xx} F_{xx} - B_{xx} E_{xx}), \\ \bar{D}_{xx}^* &= A_{xx} \bar{D}_{xx} - B_{xx} \bar{B}_{xx}, \quad D_{xx}^* = A_{xx} D_{xx} - B_{xx} B_{xx}. \end{aligned} \tag{97c}$$

Integrating the two equations in (97a) and (97b), we obtain

$$\bar{D}_{xx}^* u(x) = -\bar{B}_{xx}\left(-\int^x \int^\xi \int^\eta q(\zeta)\,d\zeta\,d\xi d\eta + K_2 \frac{x^2}{2} + K_3 x + K_4\right)$$
$$+ J_1 \frac{dw}{dx} + (P_1 + \bar{D}_{xx} K_1)x, \tag{98}$$

$$\bar{D}_{xx}^* \phi_x(x) = A_{xx}\left(-\int^x \int^\xi \int^\eta q(\zeta)\,d\zeta\,d\xi d\eta + K_2 \frac{x^2}{2} + K_3 x + K_5\right)$$
$$+ J_2 \frac{dw}{dx} + (P_2 - B_{xx} K_1)x, \tag{99}$$

where $K_4$ and $K_5$ are the constants of integration.

We return to Equation (95) and write it in terms of the generalized displacements (the differential of the constant part involving $P_1$, $P_2$, and $K_1$ is set to zero):

$$\begin{aligned} 0 &= -\hat{A}_{xz}\left(\phi_x + \frac{dw}{dx}\right) + \left(-\int^x q(\xi)\,d\xi + K_2\right) \\ &\quad - \alpha \frac{d}{dx}\left(E_{xx}\frac{du}{dx} + \bar{F}_{xx}\frac{d\phi_x}{dx} - \alpha H_{xx}\frac{d^2w}{dx^2}\right) \\ &= -\frac{\hat{A}_{xz}}{\bar{D}_{xx}^*}\left[A_{xx}\left(-\int^x \int^\xi \int^\eta q(\zeta)\,d\zeta\,d\xi d\eta + K_2\frac{x^2}{2} + K_3 x + K_5\right)\right. \\ &\quad \left. + J_2\frac{dw}{dx} + (P_2 - B_{xx}K_1)x + \bar{D}_{xx}^*\frac{dw}{dx}\right] \\ &\quad - \frac{1}{\bar{D}_{xx}^*}\frac{d}{dx}\left[P_3 F(x) + \alpha(E_{xx} J_1 + \bar{F}_{xx} J_2 - \alpha H_{xx}\bar{D}_{xx}^*)\frac{d^2w}{dx^2}\right] \\ &\quad + \left(-\int^x q(\xi)\,d\xi + K_2\right). \end{aligned} \tag{100}$$

where

$$P_3 = \alpha(A_{xx}\bar{F}_{xx} - \bar{B}_{xx}E_{xx}), \quad \hat{D}_{xx}^* = \bar{D}_{xx}^* - P_3 = A_{xx}\hat{D}_{xx} - \bar{B}_{xx}\bar{B}_{xx}. \tag{101}$$

Integrating Equation (100) once and collecting the like terms together, we obtain

$$
\begin{aligned}
0 = & -\frac{\hat{A}_{xz}}{\bar{D}^*_{xx}}(\bar{D}^*_{xx} + J_2)w + \frac{\alpha}{\bar{D}^*_{xx}}(\alpha\,\bar{D}^*_{xx}\,H_{xx} - E_{xx}\,J_1 - \bar{F}_{xx}\,J_2)\frac{d^2 w}{dx^2} \\
& -\frac{\hat{A}_{xz}}{\bar{D}^*_{xx}}\Bigg[ A_{xx}\bigg(-\int^x\int^\xi\int^\eta\int^\zeta q(\mu)\,d\mu\,d\zeta\,d\eta\,d\xi \\
& \qquad\qquad + K_2\frac{x^3}{6} + K_3\frac{x^2}{2} + K_5\,x + K_6 \bigg) + (P_2 - B_{xx}\,K_1)\frac{x^2}{2}\Bigg] \\
& +\frac{\hat{D}^*_{xx}}{\bar{D}^*_{xx}}\bigg(-\int^x\int^\xi q(\eta)\,d\eta\,d\xi + K_2 x\bigg) - \frac{P_3}{\bar{D}^*_{xx}}K_3 \\
= & -c_1\,w + c_2\frac{d^2 w}{dx^2} + g(x),
\end{aligned}
\tag{102}
$$

where

$$
\begin{aligned}
c_1 &= \frac{\hat{A}_{xz}}{\bar{D}^*_{xx}}(\bar{D}^*_{xx} + J_2) = \frac{\hat{A}_{xz}D^*_{xx}}{\bar{D}^*_{xx}}, \\
c_2 &= \frac{\alpha}{\bar{D}^*_{xx}}(\alpha\,\bar{D}^*_{xx}\,H_{xx} - E_{xx}\,J_1 - \bar{F}_{xx}\,J_2),
\end{aligned}
\tag{103}
$$

$$
\begin{aligned}
g(x) = & -\frac{\hat{A}_{xz}}{\bar{D}^*_{xx}}\Bigg[ A_{xx}\bigg(-\int^x\int^\xi\int^\eta\int^\zeta q(\mu)\,d\mu\,d\zeta\,d\eta\,d\xi + K_2\frac{x^3}{6} + K_3\frac{x^2}{2} \\
& \qquad\qquad + K_5\,x + K_6 \bigg) + (P_2 - B_{xx}\,K_1)\frac{x^2}{2}\Bigg] \\
& +\frac{\hat{D}^*_{xx}}{\bar{D}^*_{xx}}\bigg(-\int^x\int^\xi q(\eta)\,d\eta\,d\xi + K_2 x\bigg) - \frac{P_3}{\bar{D}^*_{xx}}K_3.
\end{aligned}
\tag{104}
$$

It is clear from Equation (102) that the analytical solution to the RBT is not algebraic but hyperbolic (because $c_1 > 0$ and $c_2 > 0$). The homogeneous solution of Equation (102) is

$$
w_h(x) = K_7\cosh\mu x + K_8\sinh\mu x, \quad \mu = \sqrt{\frac{c_1}{c_2}}.
\tag{105}
$$

The total solution $w(x)$ is obtained by adding the particular solution, $w_p(x)$ due to $g(x)$: $w(x) = w_h(x) + w_p(x)$. In addition, there are eight constants of integration, including the two constants of integration introduced in Equation (105). In the RBT, one is required specify $-dw/dx$ in addition to $\phi_x$ (or their dual variables, $P_{xx}$ and $\bar{M}_{xx}$, respectively), providing the required eight boundary conditions.

The second-order derivative appearing in Equation (102) comes from $P_{xx}$ of Equation (92). If we neglect the second-order derivative of $w$ in Equation (102), by reasoning that they are higher-order terms (i.e., very small compared to the other terms in the equation), we obtain

$$
\begin{aligned}
0 = & -\frac{\hat{A}_{xz}D^*_{xx}}{\bar{D}^*_{xx}}w - \frac{\hat{A}_{xz}}{\bar{D}^*_{xx}}\Bigg[ A_{xx}\bigg(-\int^x\int^\xi\int^\eta\int^\zeta q(\mu)\,d\mu\,d\zeta\,d\eta\,d\xi \\
& \qquad\qquad\qquad + K_2\frac{x^3}{6} + K_3\frac{x^2}{2} + K_5\,x + K_6 \bigg) \\
& \qquad\qquad\qquad + \Big( A_{xx}M_T^{(1)} - B_{xx}M_T^{(0)} - B_{xx}K_1 \Big)\frac{x^2}{2}\Bigg] \\
& +\frac{\hat{D}^*_{xx}}{\bar{D}^*_{xx}}\bigg(-\int^x\int^\xi q(\eta)\,d\eta\,d\xi + K_2 x\bigg) - \frac{P_3}{\bar{D}^*_{xx}}K_3,
\end{aligned}
\tag{106}
$$

The solution to these equations is (after a lengthy algebra)

$$u(x) = \frac{1}{A_{xx}}\left(N_{xx}^T + K_1\right)x, \tag{107}$$

$$\phi_x(x) = \frac{1}{D_{xx}}\left(-\int^x \int^\xi \int^\eta q(\zeta)\,d\zeta\,d\xi d\eta + K_2\frac{x^2}{2} + K_3 x\right)$$
$$+ \frac{17}{56}\frac{1}{A_{xz}}\left(-\int^x q(\xi)\,d\xi + K_2\right) + \frac{1}{D_{xx}}M_{xx}^T x + \frac{1}{D_{xx}}K_5, \tag{108}$$

$$w(x) = -\frac{1}{D_{xx}}\left(-\int^x \int^\xi \int^\eta \int^\zeta q(\mu)\,d\mu\,d\zeta\,d\eta\,d\xi + K_2\frac{x^3}{6} + K_3\frac{x^2}{2}\right.$$
$$\left. + K_5 x\right) - \frac{1}{D_{xx}}M_{xx}^T\frac{x^2}{2} - \frac{1}{D_{xx}}K_6 - \frac{2}{7}\frac{1}{A_{xz}}K_3$$
$$+ \frac{17}{14}\frac{1}{A_{xz}}\left(-\int^x \int^\xi q(\eta)\,d\eta d\xi + K_2 x\right), \tag{109}$$

The corresponding solutions obtained using the first-order shear deformation theory (TBT) of beams are:

$$u(x) = \frac{1}{A_{xx}}\left(N_{xx}^T + K_1\right)x, \tag{110}$$

$$\phi_x(x) = \frac{1}{D_{xx}}\left(-\int^x \int^\xi \int^\eta q(\zeta)\,d\zeta\,d\eta d\xi + K_2\frac{x^2}{2} + K_3 x\right)$$
$$+ \frac{1}{D_{xx}}M_{xx}^T x + \frac{1}{D_{xx}}K_5, \tag{111}$$

$$w(x) = -\frac{1}{D_{xx}}\left(-\int^x \int^\xi \int^\eta \int^\zeta q(\mu)\,d\mu\,d\zeta\,d\eta d\xi + K_2\frac{x^3}{6}\right.$$
$$\left. + K_3\frac{x^2}{2} + K_5 x\right) - \frac{1}{D_{xx}}M_{xx}^T\frac{x^2}{2} - \frac{1}{D_{xx}}K_6$$
$$+ \frac{1}{A_{xz}}\left(-\int^x \int^\xi q(\eta)\,d\eta\,d\xi + K_2 x\right), \tag{112}$$

A comparison of Equation (112) with Equation (109) shows that the shear correction factor predicted by the "simplified" RBT is $K_s = 14/17$, which is slightly smaller than the value suggested for rectangular cross-section beams, which is $K_s = 5/6$. Of course, the TBT solution is different from the RBT solution. Particularly, the expression for $\phi_x$ in the RBT contains a term due to transverse shear coefficient.

As an example, we present here the exact solutions using the simplified RBT of a functionally graded beam with both ends *pinned* and subjected to a uniformly distributed load of magnitude $q_0$. We take the origin of the $x$-coordinate at the left end of the beam (i.e., $0 \leq x \leq L$). For this case, the boundary conditions at both ends, $x = 0$ and $x = L$, are:

$$u = w = 0, \quad \bar{M}_{xx} = 0 \rightarrow M_{xx} = 0. \tag{113}$$

Use of these boundary conditions yield

$$u(x) = \frac{B_{xx}}{D_{xx}^*} \frac{q_0 L^3}{12} \frac{x}{L} \left(1 - 3\frac{x}{L} + 2\frac{x^2}{L^2}\right) + \frac{J_1 P_3}{\hat{A}_{xz} D_{xx}^*} \frac{q_0 L}{2} \left(1 - \frac{x}{L}\right),$$ (114)

$$\phi_x(x) = -\frac{A_{xx}}{D_{xx}^*} \frac{q_0 L^3}{12} \frac{x}{L} \left(1 - 3\frac{x}{L} + 2\frac{x^2}{L^2}\right) - \frac{1}{D_{xx}} \frac{q_0 L^3}{24} \left(1 - 2\frac{x}{L}\right)$$
$$- \frac{M_{xx}^T L}{2D_{xx}} \left(1 - 2\frac{x}{L}\right) + \frac{J_2 P_3 q_0}{2D_{xx}^* \hat{A}_{xz}} \left(2J_2 + J_1 \frac{B_{xx}}{D_{xx}}\right) \left(1 - 2\frac{x}{L}\right),$$ (115)

$$w(x) = \frac{A_{xx}}{D_{xx}^*} \frac{q_0 L^4}{24} \frac{x}{L} \left(1 - 2\frac{x^2}{L^2} + \frac{x^3}{L^3}\right) - \frac{B_{xx}^2}{D_{xx} D_{xx}^*} \frac{q_0 L^4}{24} \frac{x}{L} \left(1 - \frac{x}{L}\right)$$
$$+ \frac{M_T^{(1)} L^2}{2D_{xx}} \frac{x}{L} \left(2 - \frac{x}{L}\right) + \frac{P_3 q_0 L^2}{2\hat{A}_{xz} D_{xx}^*} \left(2 - J_1 \frac{B_{xx}}{D_{xx}}\right) \frac{x}{L} \left(1 - \frac{x}{L}\right).$$ (116)

## 5. Summary

In this paper three different beam theories, namely, the classical, first-order, and third-order beam theories are presented for beams, accounting for the through-thickness variation of the material, modified couple stress effect, and the von Kármán nonlinearity. Exact solutions for bending of the three theories are presented for several boundary conditions.

Numerical examples are also presented to illustrate the accuracy of various models and bring out certain salient features of functionally graded beams. A study of the FGM beams also revealed that the dimensionless bending deflections ($\bar{w} = w\,\hat{D}_{xx}/q_0 L^4$) are not monotonic functions of the power-law index/exponent $n$ because the coupling stiffness $B_{xx}$ is not a monotonically increasing or decreasing function of the modulus ratio.

Finite element models of the nonlinear theories presented herein can be found in the monograph by Reddy [34], which also contains detailed discussions of obtaining analytical and numerical solutions. A companion paper on FGM circular plates will appear following the publication of this paper (see Reddy, et al. [40]). Extensions of the theories presented herein to buckling and vibration [34,41–43], and to account for nonlocal effects [44], are also awaiting. Extension of the theories and solutions to curved beams is another major topic for interested researchers.

**Author Contributions:** Conceptualization, J.N.R.; methodology, J.N.R., E.R. and J.A.L.; software, J.N.R.; validation, J.N.R. and E.R.; formal analysis, J.N.R.; investigation, all authors; resources, J.N.R. and A.M.A.N.; data curation, J.N.R.; writing—original draft preparation, J.N.R.; writing—review and editing, all authors; visualization, J.N.R.; supervision, J.N.R. and A.M.A.N.; project administration, all authors. All authors have read and agreed to the published version of the manuscript.

**Funding:** This research received no external funding.

**Institutional Review Board Statement:** Not applicable.

**Informed Consent Statement:** Not applicable.

**Acknowledgments:** J. N. Reddy gratefully acknowledges the support of this research by the O'Donnell Foundation Chair IV at Texas A&M University and the Distinguished Professorship at University of North Texas, Denton. E. Ruocco acknowledges the financial support "VALERE: VAnviteLli pEr la RicErca" by University of Campania Luigi Vanvitelli.

**Conflicts of Interest:** The authors declare no conflict of interest.

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
