# Peer review of "Theories and Analysis of Functionally Graded Beams"

_applsci, doi:10.3390/app11157159_

Round 1
Reviewer 1 Report
In this paper, the authors conducted a comprehensive literature survey related to the classical and the first- and higher-order shear deformation theories of functionally graded (FG) beams, for which the governing equations and analytical solutions were presented. The modified couple stress effects and the von Karman nonlinearity effects were also considered in the formulation of this paper.
Some suggestions are given as follows:
- The authors may explain the difference between the boundary conditions of the FG beams in the modified couple stress beam theories and those in the classical continuum mechanics-based beam theories.
- The authors may explain how to obtain the material length scale parameter (l) when the couple stress effects were considered.
- The paper is well organized and well written. I strongly recommend that it should be accepted by Applied Sciences. The above suggestions are given for the authors’ reference, and the second run of review is no need.
Author Response
Dear Editor,
thank you very much for your email and for the opportunity to resubmit the paper in a revised version that incorporates/receives the Reviewer(s) suggestions. We examined in detail the Reviewers' comments and we feel that all the points raised by them could be addressed to both better explain some aspects of the work and clarify all the issues at hand.
Below, all the replies to the Reviewers' comments are point-by-point reported, by highlighting – as requested – all the changes/amendments considered in the revised version of the manuscript (in black the original Reviewers comments/questions, in red the Authors responses).

Reviewer 2 Report
In the present paper, under the title "Theories and Analysis of Functionally Graded Beams", the authors present a work that focuses on the governing equations and analytical solutions of the classical and shear deformation theories of functionally graded straight beams. The work outlines the displacement fields of the three theories (classical, first-order, and third-order), the governing equations, and analytical solutions of straight beams for the linear case.
The article is very well organized and presents in a very clear and well-founded way the different theories of beams.
The work has a great scientific relevance and is robustly based on works that are fundamental references for the methodologies that are explained.
The examples presented clearly demonstrate the strengths and limitations of the different theories and boundary conditions used. The results presented are relevant and can be useful for future comparisons' purpose. The discussion carried out is well supported and the conclusions clearly characterized the work.
There are small details that we propose to be taken care of at work.
So at the beginning of line 17, where it is “smaller”, it should be exactly the opposite, that is, “bigger”.
In equation 7 the mathematical formulation presented must be explained.
Although it is stated that the article is not a review work on the subject presented as stated in the work, the indication of review works published in 2007 and 2010 should be complemented with at least one much more recent reference (2020 or 2021 ).
The discussion carried out is well supported and the conclusions clearly characterized the work carried out.
In conclusion, my overall opinion is that the work is well exposed, presents a topic with scientific relevance and it may be very useful for further studies and after correcting the small flaws we indicated, the article may be considered for publication in the Journal Applied Sciences.
Author Response

(The authors gave the same response as above.)
